# Prognostic Value of the Immunohistochemical Detection of Cellular Components of the Tumor Microenvironment in Oral Squamous Cell Carcinoma: A Systematic Review

**DOI:** 10.3390/cimb47070544

**Published:** 2025-07-12

**Authors:** Hannah Gil de Farias Morais, Caroline Fernandes da Costa, Maurília Raquel de Souto Medeiros, Bárbara de Assis Araújo, Everton Freitas de Morais, Ricardo D. Coletta, Roseana de Almeida Freitas

**Affiliations:** 1Department of Oral Pathology, Federal University of Rio Grande do Norte, Natal 59056-000, RN, Brazil; hannah.gil@ufrn.br (H.G.d.F.M.); carolfc1602@gmail.com (C.F.d.C.); mauriliaraquel@hotmail.com (M.R.d.S.M.); dbarbara.assis@gmail.com (B.d.A.A.); roseanafreitas@hotmail.com (R.d.A.F.); 2Multicampi School of Medical Sciences, Federal University of Rio Grande do Norte, Caicó 59300-000, RN, Brazil; 3Piracicaba Dental School, University of Campinas (UNICAMP), Piracicaba 13414-903, SP, Brazil; coletta@fop.unicamp.br

**Keywords:** oral squamous cell carcinoma, oral cancer, tumor microenvironment, prognosis, immunohistochemistry

## Abstract

This study aims to investigate the prognostic impact of cellular components of the tumor microenvironment (TME), analyzed through immunohistochemistry, in oral squamous cell carcinoma (OSCC). This review was conducted following the guidelines of the Preferred Reporting Items for Systematic Reviews and Meta-Analyses (PRISMA). Searches were performed in EMBASE, Medline/PubMed, Cochrane Collaboration Library, Web of Science, ScienceDirect, Scopus, and Google Scholar. After applying the study criteria, 59 articles were included, involving the analysis of cancer-associated fibroblasts (CAFs), immune cells, and endothelial cells. It was found that TME rich in α-SMA-positive CAFs, tumor-associated macrophages, and dendritic cells contribute to the invasion and progression of OSCC, resulting in a poorer prognosis. In contrast, the presence of high amounts of NK CD57^+^ cells, CD8^+^/CD45RO^+^ T cells, and PNAd^+^ endothelial cells are associated with anti-tumor immune responses in OSCC and improved survival rates. CD3^+^ and CD4^+^ T cells, Treg cells, B cells, and mast cells have shown little to no evidence of prognostic utility. Several stromal components of TME were found to have a strong impact on the aggressiveness of OSCC, reaffirming the potential use of these biomarkers as prognostic tools and therapeutic targets.

## 1. Introduction

Squamous cell carcinoma is the most common malignant neoplasm of the oral cavity and constitutes a major global public health issue due to its high morbidity and mortality rates [1,2]. Head and neck cancers are typically staged using the Tumor-Node-Metastasis (TNM) system established by the American Joint Committee on Cancer (AJCC), which considers the extent of the primary tumor (T), regional lymph node involvement (N), and the presence of distant metastases (M) [3]. Although this system provides a reliable framework for prognostic stratification and therapeutic planning, some cases of oral squamous cell carcinoma (OSCC) still progress with local recurrence or metastasis, even when diagnosed at an early stage and properly treated [4]. Treatment is generally determined based on clinical staging, with primary surgery, with or without adjuvant chemoradiotherapy, being the main approach recommended by the National Comprehensive Cancer Network (NCCN) guidelines [5]. With the increasing incidence of oral cancer in recent decades, research efforts have intensified toward the identification of more accurate prognostic biomarkers, protein expression patterns, and oncogenic mutations in OSCC, aiming to enhance the biological understanding of the disease and improve risk stratification [6].

In recent years, it has become increasingly evident that stromal adaptations facilitate neoplastic dissemination, reinforcing the critical role of tumor microenvironment (TME) alterations in tumor progression [7]. Tumors are no longer viewed merely as masses of malignant cancer cells, but rather as complex ecosystems in which various non-malignant cell subpopulations are recruited to form a self-sustaining biological structure [8]. As a result, recent studies have placed growing emphasis on understanding the fundamental role of the TME in regulating cancer development, including in OSCC [9].

The stromal tumor–cell complex includes neoplastic cells, blood vessels, extracellular matrix, inflammatory cells, endothelial cells, fibroblasts, and a specific subtype of fibroblast known as cancer-associated fibroblast (CAF), which together can critically influence the carcinogenesis process, either through cell–cell contact or the secretion of cytokines [10]. Evidence has indicated that immune infiltration of the TME, intratumoral heterogeneity, and interactions between malignant and non-malignant cells are critical for various aspects of tumor biology [11,12].

Despite the growing number of studies on the TME over the past decade, understandings of its dynamics remain fragmented, particularly concerning the prognostic value of its diverse cellular components in OSCC [13]. Previous reviews have often focused on specific cell types or lacked a comprehensive synthesis of immunohistochemically assessed markers within the TME. In this context, the present systematic review provides an updated and integrated analysis of the main cellular elements of the TME—including CAFs, immune cells, and endothelial cells—and their prognostic significance in OSCC. By consolidating findings from multiple studies, our review offers a comparative perspective on how these components influence tumor progression and patient survival, thereby filling a critical gap in the literature and reinforcing the potential of these markers as prognostic tools and therapeutic targets.

## 2. Materials and Methods

This systematic review followed the Preferred Reporting Items for Systematic Reviews and Meta-Analyses (PRISMA) guidelines [14]. The PICOS criteria were used to develop the research questions: (i) population—patients diagnosed with OSCC; (ii) intervention—immunohistochemical analysis of the cellular components of OSCC TME; (iii) control group—not applicable; (iv) outcome—overall survival (OS), disease-free survival (DFS), and disease-specific survival (DSS); (v) study design—observational studies in humans. The PRISMA checklist is available in Appendix A.

The research question was: “Does the immunohistochemical analysis of cellular components of the tumor microenvironment in OSCC have predictive value in the prognosis of patients?” The study protocol was registered in PROSPERO under the number CRD42024506839.

### 2.1. Search Strategy

To identify primary research articles that assessed the prognostic impact of immunohistochemical markers in OSCC, we searched the databases EMBASE, Medline/PubMed, Cochrane Collaboration Library, Web of Science, ScienceDirect, Scopus, and Google Scholar. Additionally, the references of pre-selected articles were manually searched. The searches were conducted in April 2024. A second literature search was performed in early September 2024, retrieving articles published between May and August 2024.

The cellular components of the tumor microenvironment stroma in OSCC were defined as established by Anderson and Simon [15], including T cells, B cells, natural killer (NK) cells, macrophages, neutrophils, dendritic cells, endothelial cells, cancer-associated fibroblasts (CAFs), and adipocytes. Based on this, eight search strategies were defined, based on combinations of keywords involving all the cellular components addressed in this review. The full search strategies are provided in Appendix A.

### 2.2. Study Selection and Selection Criteria

The initial screening based on titles and abstracts was conducted by four independent reviewers who classified the studies as “yes,” “no,” or “maybe” based on the inclusion criteria. Studies classified as “yes” or “maybe” were selected for full-text reading. The eligibility criteria were applied at both stages.

Articles that evaluated the predictive impact of tumor microenvironment cellular components in OSCC, analyzed by immunohistochemistry, were included in this systematic review. The search was conducted with no time or language restrictions. The following studies were excluded from this systematic review: (i) in vitro studies and/or those using only animal models and/or peripheral blood samples; (ii) review articles and editorials; (iii) studies that did not evaluate cellular components by immunohistochemistry; (iv) studies that did not assess the impact of immunohistochemical evaluation on the survival of patients with OSCC; (v) studies that included samples from the lip and oropharynx; (vi) studies that utilized only in silico or bioinformatics analyses or performed multiplex or dual immunohistochemistry staining. Four reviewers independently selected the articles, and any discrepancies were resolved by consensus. Figure 1 illustrates the flowchart of the study screening and selection process.

### 2.3. Data Extraction

The authors independently extracted the following data from the included studies using a pre-established form: authors, year of publication, country, study type, sample size, patient allocation method, tumor sublocation in the oral cavity, sex, age, TNM staging, tumor size, lymph node and distant metastasis, local recurrence, mortality, histological grading, follow-up time, analyzed cell, biomarker used, and immunohistochemical analysis method. The results of the individual studies were then summarized, categorized, compiled, and analyzed.

### 2.4. Quality Assessment and Risk of Bias

The Reporting Recommendations for Tumor Marker Prognostic Studies (REMARK) guidelines [16,17] were used to assess the quality of the included studies. The risk of bias in the studies was analyzed according to a checklist based on the Meta-Analysis of Statistics Assessment and Review Instrument (MAStARI) [18]. Four reviewers answered 9 questions for descriptive studies with Y for “yes,” N for “no,” U for “unclear,” and NA for “not applicable.” Disagreements were resolved through discussion among the four authors.

## 3. Results and Discussion

A total of 797 articles were retrieved. After the removal of duplicates, 273 articles remained for the first screening based on titles and abstracts. One hundred and nineteen articles were selected for the next phase. After reading the full text, 59 articles were included in the qualitative synthesis [19,20,21,22,23,24,25,26,27,28,29,30,31,32,33,34,35,36,37,38,39,40,41,42,43,44,45,46,47,48,49,50,51,52,53,54,55,56,57,58,59,60,61,62,63,64,65,66,67,68,69,70,71,72,73,74,75,76,77]. The list of excluded studies and the reasons for exclusion are shown in Appendix A. The selected studies were all retrospective cohort studies, with non-randomized patient allocation, and were published between 1998 and 2023. The studies were conducted in 13 different countries, 20 (33.9%) of which were by Chinese research groups.

Regarding methodological characteristics, all articles included in this systematic review involved 5200 patients, with an average of 87 participants per study. Our analysis revealed that 2129 (52.3%) of the cases assessed were lesions diagnosed at early stages according to clinical TNM staging, when such information was available. Among these cases, 87.9% were classified as well or moderately differentiated according to the WHO histopathological classification.

Standard therapies for OSCC include surgical excision, radiation therapy, and chemotherapy, which can be administered alone or in combination, depending on the stage of the disease. Although advances in these treatments have improved survival rates for early-stage OSCC, patients with advanced cancer still face discouraging outcomes, with a five-year survival rate of less than 50% [11]. Moreover, treatment resistance, local recurrence, and distant metastasis are significant therapeutic challenges that require a deeper understanding of OSCC biology and progression. Therefore, a comprehensive understanding of the interactions between the parenchyma and stroma within the complex tumor microenvironment of OSCC is crucial for developing promising therapeutic strategies that could positively impact the prognosis of oral cancer.

In this systematic review, we thoroughly examined and summarized the relationship between biomarkers evaluated through immunohistochemical techniques and their association with clinical prognostic parameters and survival outcomes. We chose to include only studies that used immunohistochemistry due to its practical and economic advantages. Immunohistochemistry is widely used in clinical practice and biomedical research because it is an accessible technique with relatively low costs compared to more advanced methods. Based on our systematic review of 59 articles involving 5200 patients, we observed that various stromal biomarkers of the tumor microenvironment (TME) have a strong impact on the aggressiveness profile of OSCC. Figure 2 represents the TME in OSCC, with the stromal cellular components analyzed in this systematic review, their immunohistochemical markers, and the markers that showed the greatest impact on the survival of patients diagnosed with OSCC.

### 3.1. Cancer-Associated Fibroblasts

Among the potential prognostic biomarkers for OSCC, we highlight those associated with the identification of cancer-associated fibroblasts (CAFs). CAFs, one of the most active components in regulating tumor biology, function as “factories” that produce a wide range of molecular factors. These factors enhance tumor cell proliferation, remodel the extracellular matrix, alter the metabolic properties of tumor cells, reprogram the anti-tumor immune response, facilitate metastasis, and contribute to therapy resistance [78,79].

Within the selected studies, 15 [19,24,46,47,48,49,50,51,52,53,56,58,59,63,77] analyzed the impact of CAF evaluation on the prognosis of patients diagnosed with OSCC, with 50% of the studies focusing solely on tumors located in the oral tongue [46,47,48,49,52,56,59]. For the identification of CAFs, the studies used the α-SMA marker, with some studies complementing the analysis with TGF-β [56,63] and FAP [24]. These studies demonstrate that a tumor microenvironment rich in α-SMA-positive CAFs has an unfavorable impact on cancer invasion and progression, the risk of locoregional recurrence, occult and distant metastases, and increased mortality in patients with OSCC, confirming the critical role of CAFs in tumor development. The main characteristics, immunohistochemical evaluation methods, and findings of the studies included in this systematic review that assessed CAFs are shown in Appendix A.

In our systematic review, α-SMA was a biomarker strongly associated with worse prognosis in OSCC, particularly in OTSCC. Bello et al. [49] highlighted that a tumor microenvironment rich in α-SMA-positive CAFs is associated with increased disease-specific mortality in OTSCC. Vered et al. [59] further emphasized the unfavorable impact on the risk of recurrence, particularly in young patients. Li et al. [48] and Vered et al. [47] confirmed the independent prognostic value of α-SMA-positive CAFs in OTSCC. These findings reveal the potential of identifying CAFs through α-SMA immunoexpression to predict survival in patients diagnosed with OTSCC.

High α-SMA expression has also been suggested to be associated with the presence of occult cervical metastases and chemotherapy resistance in OSCC. Liang et al. [50] observed an increase in stromal α-SMA expression after chemotherapy in OSCC samples, suggesting that chemotherapy may increase CAFs in these tumors and that this could be a factor associated with chemotherapy resistance in certain cancers. Luksic et al. [77] highlighted a highly significant association with the presence of occult metastasis in the neck and with patient survival. These results suggest that a high expression of α-SMA-positive CAFs could be a biomarker to identify patients more likely to exhibit chemotherapy resistance and those who could benefit from elective cervical dissection.

Previous studies have revealed significant associations between high densities of α-SMA-positive CAFs in the tumor microenvironment of OSCC and poor patient survival [80]. In contrast, there was a complete lack of consistency in the evaluation of CAF density across the studies included in this systematic review. Nevertheless, this did not prevent the association between high CAF density and poor prognosis, suggesting that regardless of the methodology used to assess density, its biological impact on prognosis remained evident. Despite these findings, it is crucial to develop a standardized and validated immunohistochemical analysis method aimed at universalizing CAF identification biomarkers as potential prognostic markers for OSCC.

### 3.2. CD163+/CD68+ Macrophages, CD1a+ Dendritic Cells (DCs), CD57+ Natural Killer (NK) Cells, and CD8+/CD45RO+ T Cells Have Prognostic Potential

Previous research has shown that the immune system is involved in the development and progression of OSCC [81]. The tumor immune microenvironment plays an ambivalent role, potentially recognizing tumor cells and inhibiting tumor development, but it can also contribute to immune escape and promote a pro-tumoral inflammatory response [82,83]. These findings align with those observed in our systematic review. Indeed, more important than simply identifying an immune response within the tumor microenvironment is understanding which components of this immune response are present and actively involved. Given that the immune status is closely related to OSCC, immunotherapy for OSCC has garnered significant attention. The main goal of cancer immunotherapy is to activate the damaged immune system by stimulating anti-tumor immune cells and negatively regulating immunosuppressive cells. In 2016, PD-1-targeted drugs (Pembrolizumab) and anti-PD-1 monoclonal antibodies (Nivolumab) were approved by the U.S. Food and Drug Administration (FDA) to treat patients with recurrent or metastatic OSCC [84,85]. Therefore, a deeper understanding of tumor immune microenvironment biomarkers could help identify (1) potential new prognostic markers, (2) promising therapeutic targets, and (3) the development of biomarker panels that can be used to screen patients and guide them toward individualized immunotherapy based on the characteristics of their tumor’s immune microenvironment.

Macrophages were one of the most extensively studied cells identified in this systematic review [21,29,30,56,57,67,76], with the majority of studies reporting a negative effect of macrophages on the survival of patients diagnosed with OSCC, with survival outcomes differing according to the analyzed compartment. Multivariate analyses revealed that a high density of CD163-positive macrophages at the tumor invasion front is an independent predictive factor for DFS in patients with OSCC [76], while a large number of CD68-positive intratumoral macrophages are independent prognostic negative markers for overall survival (OS) [21,57].

Peritumoral and tumor-associated CD1a-positive dendritic cells (DCs) have been shown to impact both DFS^31^ and OS [38,42] in patients with OSCC, with the depletion of peritumoral CD1a+ cells proving to be an independent factor associated with OS and DFS [38]. The presence of plasmacytoid dendritic cells (pDCs), CD123+, was also significantly associated with reduced OS [39,40]. Multivariate analysis confirmed that high levels of tumor-infiltrating pDCs were an independent prognostic factor [39]. These results may suggest an association between the accumulation of DCs and functional immune impairment.

Univariate regression analyses showed that a higher expression of CD57-positive natural killer (NK) cells was positively correlated with improved OS [30]. The results confirmed by multivariate analyses demonstrated that CD57 expression was an independent prognostic factor for 5-year OS [30,68]. It has also been demonstrated that high infiltration of CD57+ NK cells indicates favorable OS in early-stage OSCC [32]. These results suggest the potential utility of investigating CD57 immunolabeling to predict the prognosis of OSCC, particularly in early-stage cases.

Most studies evaluating the CD8-positive T cell subset revealed that higher immunoexpression of these cells was positively correlated with improved OS [26,27,30,32,68]. Li et al. [68] and Shimizu et al. [27] also confirmed these findings in multivariate analyses, with CD8 expression at the tumor invasion front identified as an independent prognostic factor for 5-year OS and recurrence-free survival (RFS). These results strongly suggest that the assessment of CD8+ T cells at the invasion front provides an indicator of recurrence and tumor prognosis in OSCC.

Only one study included in this review investigated the impact of CD45RO evaluation in OSCC [28], revealing promising results, with cases showing high expression of this marker significantly associated with higher rates of DFS and OS both at the tumor center and the invasion front. A high expression of CD45RO^+^ at the invasion front demonstrated a significant correlation with DFS in multivariate analysis, and with both OS in the tumor center and the front.

### 3.3. CD3+ and CD4+ T Cells, Tregs, B Cells, and Mast Cells Do Not Show Evidence of Prognostic Utility, or Such Evidence Is Scarce

For the T cell subsets, only one study demonstrated that a high number of CD3^+^ T cells was an independent prognostic marker of better OS in OTSCC [29]. Other studies failed to find statistically significant associations between CD3 and OS or DFS in patients with OSCC [28,55]. None of the studies demonstrated an impact of CD4+ stromal T cells on the survival of patients with OSCC [26,30,75].

The studies included in this systematic review that evaluated FOXP3^+^ regulatory T cells (Tregs) presented contradictory results. Liang et al. [31], Quan et al. [74], and Dayan et al. [56] reported that high expression of FOXP3 was inversely associated with OS and DFS in patients with OSCC. Liang et al. [31] also demonstrated through multivariate analysis that FOXP3 expression was an independent prognostic indicator for OTSCC. In contrast to these findings, Boxberg et al. [61], Koike et al. [69], and Lequerica-Fernandez et al. [33] observed that the impact of FOXP3+ Tregs was directly proportional to patient survival, with Lequerica-Fernandez et al. [33] showing an association with DSS and Koike et al. [69] with OS, DSS, and DFS at five years at the tumor invasion front. These results suggest that the tumor type and other components of the TME may influence a Treg profile with either anti-tumor or immunosuppressive effects.

The vast majority of studies investigating B cells demonstrated through univariate analyses that high CD20 expression is predictive of better OS [29,32,57]. Huang et al. [32] also observed that high B cell CD20+ infiltration indicates favorable OS in early-stage OSCC. Other B cell markers were also studied, with positive impact results on OS, such as CD19 [52] and CD138 [74]. However, it is important to highlight that none of the studies included confirmed this impact in multivariate analyses.

Only one study evaluating mast cells was included in this systematic review [62]. It used tryptase as a cellular marker and included only OTSCC samples, but the results were not statistically significant, showing no correlations with survival rates. The main characteristics, immunohistochemical evaluation methods, and findings of the studies included in this systematic review that assessed immune microenvironment markers are shown in Appendix A.

### 3.4. Angiogenesis

Our systematic review also highlighted the role of angiogenesis, a key process in the tumorigenesis of solid cancers, promoting tumor growth, recurrence, and metastasis in OSCC. Angiogenesis (or neoangiogenesis, the formation of new blood vessels) is crucial for tumor growth, invasion, and metastasis. It is primarily mediated through the vascular endothelial growth factor (VEGF) pathway [86]. The U.S. Food and Drug Administration (FDA) has approved several anti-angiogenic agents for treating solid tumors such as colorectal cancer, renal cell carcinoma, ovarian cancer, gastric cancer, and thyroid cancer [86]. Contrary to concerns that angiogenesis inhibitors might increase hypoxia and lead to treatment resistance, preclinical models suggest these inhibitors can overcome resistance and synergize with traditional therapies, such as radiation [87].

Although recognized as important components of the tumor microenvironment in malignant neoplasms, studies analyzing the impact of angiogenesis markers on the prognosis of OSCC through immunohistochemistry are scarce. Seven studies were included, analyzing the following markers: CD3466, PNAd39,51, vWF [54,71], D2-40 [54,57,62,71,73], and LYVE-1 [46].

The studies revealed conflicting results regarding the impact of lymphangiogenesis on the survival of patients with OSCC. Chung et al. [72] identified through multivariate analyses that intratumoral lymphatic density analysis, using D2-40 and CD34, was the only independent predictive variable correlated with regional metastases in OTSCC. Supporting these findings, Mafra et al. [62] observed that intratumoral lymphatic density (D2-40) was higher in advanced clinical stages (III/IV) compared to early stages, and in metastatic cases compared to non-metastatic tumors. Zhao et al. [54], also investigating lymphangiogenesis using D2-40, found similar results. On the other hand, Ding et al. [46] and Seppala et al. [71] did not identify a correlation between lymphatic vessel density and diameter with OS in patients with OTSCC. These results suggest that lymphangiogenesis indeed occurs in OSCC and that intratumoral D2-40-positive lymphatic density may serve as an indicator to infer disease aggressiveness, assess the state of lymphatic metastasis, and distinguish patients at higher risk of adverse clinical outcomes; however, further studies are required to evaluate the impact on survival in these patients.

The prognostic value of high endothelial venules was also investigated in this study through PNAd, being identified as an important marker of an anti-tumoral immune response in OSCC. Wirsing et al. [45] demonstrated that the presence of tumor-associated high endothelial venules was associated with better DSS in multivariate regression analyses. Subsequently, Wirsing et al. [57] confirmed PNAd as a marker of a favorable anti-tumoral immune microenvironment in OSCC, indicating better prognosis compared to other immune infiltrate subsets. These findings highlight that the absence of these vessels in advanced-stage tumors may identify patients with more aggressive disease and that evaluating the presence of tumor-associated high endothelial venules may help tailor the treatment of oral cancer patients to their individual needs. The main characteristics, immunohistochemical evaluation methods, and findings of the studies included in this systematic review that assessed endothelial cell markers are shown in Appendix A.

As noted in our systematic review, studies focusing on angiogenic biomarkers in OSCC using immunohistochemistry are still limited. Moreover, the results regarding their prognostic value are conflicting. It is important to highlight that anti-angiogenic agents have shown considerable toxicity and limited clinical success in patients with head and neck squamous cell carcinoma [88]. Therefore, more data is needed to clarify the true prognostic value of angiogenesis markers in OSCC, particularly in identifying patients who may benefit from anti-angiogenic treatment based on screening through such biomarkers using immunohistochemical analysis.

### 3.5. Quality Assessment and Risk of Bias

We employed the criteria established by the REMARK guidelines to evaluate the studies included in this systematic review (Appendix A). The vast majority of the selected studies had a well-defined sample (98.3%), provided details about the characteristics of the patients included in the sample (67.8%), described the analysis methods used (98.3%), defined survival analysis endpoints (74.6%), utilized appropriate statistical analyses (52.5%), and adequately reported the relationship between TME markers and patient survival in the analyzed cases (81.3%). Based on the MAStARI evaluation, most of the included studies (40.7%) were classified as having a low risk of bias (Appendix A), with 23.7% of the studies classified as having a high risk of bias, primarily due to unreliable measurement of outcomes and the use of inappropriate statistical analyses.

### 3.6. Limitations and Challenges

This systematic review provides valuable insights into the prognostic significance of stromal components in oral squamous cell carcinoma, establishing a strong foundation for advancing both research and clinical practice in this field. However, the translation of these findings into clinical application faces significant challenges, including the scarcity of in vivo functional validations, the lack of standardization in detection techniques, and the need for longitudinal studies to confirm the robustness and reproducibility of proposed markers across diverse populations. Future research should prioritize the development and validation of standardized protocols for immunohistochemical analysis of stromal biomarkers and encourage multicenter studies involving larger, more representative, and clinically well-characterized cohorts. Functional studies investigating the biological mechanisms underlying the interactions between tumor stroma and cancer cells are also critical to elucidate the biological relevance of these markers and their role in therapeutic response, particularly in the context of immunotherapy. Strengthening the integration of basic, translational, and clinical research will thus be essential for translating current knowledge into more effective diagnostic and therapeutic approaches for managing oral cancer.

Despite its comprehensive scope, several limitations were identified in the included studies. A notable lack of methodological standardization in the evaluation of stromal biomarkers—particularly CAFs and immune components—was evident, potentially introducing variability in prognostic assessments. Additionally, analyses using REMARK and MAStARI criteria revealed inconsistencies in study design, with several studies exhibiting moderate risks of bias due to incomplete reporting of confounding factors, patient allocation, and follow-up methods. These limitations hindered the conduction of a meta-analysis that could have provided clearer clarification of the study’s results. Moreover, many studies suffered from small sample sizes, limited follow-up periods, and inadequate consideration of confounding variables, all of which may affect the robustness of their findings.

In addition to the immunohistochemically validated markers analyzed in this review, recent studies have identified novel TME biomarkers with promising prognostic and therapeutic potential in OSCC. Notably, nicotinamide N-methyltransferase (NNMT) has emerged as a critical regulator of tumor–stroma interactions and cellular metabolism. NNMT modulates the activity of several epigenetic enzymes, such as sirtuins and histone deacetylases, thereby contributing to tumor progression and immune evasion [89,90]. Its overexpression has been demonstrated in various solid tumors, including OSCC, where it promotes tumor aggressiveness and poorer prognosis [91]. Importantly, several NNMT inhibitors are under investigation and have shown promise as potential agents for targeted therapy in preclinical models [92,93]. Other recently reported biomarkers include barrier-to-autointegration factor 1 (BANF1) and pescadillo ribosomal biogenesis factor 1 (PES1), both associated with immune cell infiltration and unfavorable outcomes in head and neck squamous cell carcinoma [94,95]. However, the current evidence for many of these novel markers is primarily based on transcriptomic data and bioinformatic analyses derived from public datasets. Further in vitro and in vivo validation, particularly at the protein level using immunohistochemistry, is essential to confirm their utility as prognostic indicators and therapeutic targets in OSCC.

## 4. Conclusions

This systematic review highlights the significant prognostic value of stromal components in the tumor microenvironment of oral squamous cell carcinoma (OSCC). The findings underscore the critical role of these elements in influencing tumor progression, patient survival, and therapeutic resistance. While the heterogeneity in methodologies across studies presents challenges, the consistent associations observed reaffirm the potential of stromal biomarkers as prognostic tools and therapeutic targets. This review serves as a pivotal step in bridging current knowledge with the ongoing quest for improved prognostic and therapeutic strategies in oral cancer.

## Figures and Tables

**Figure 1 cimb-47-00544-f001:**
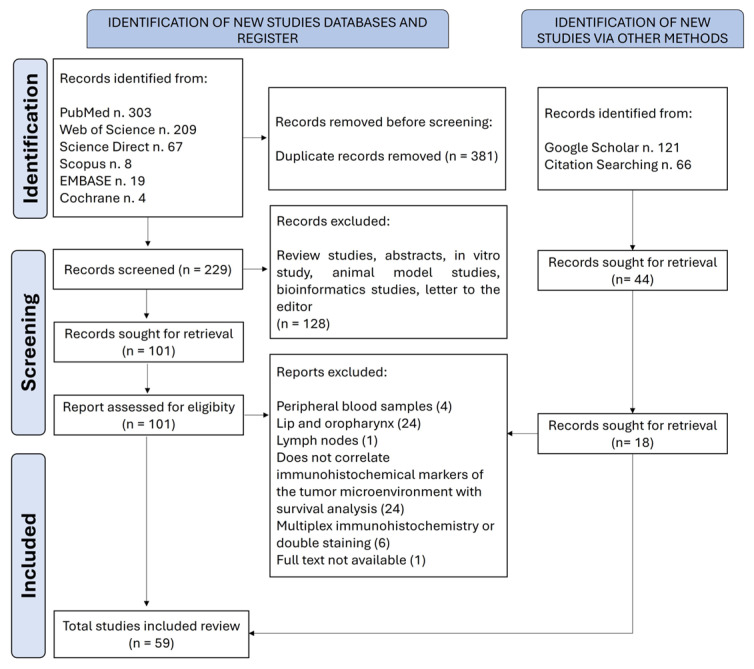
A flowchart of the literature search and selection criteria. The study followed the guidelines of the Preferred Reporting Items for Systematic Reviews and Meta-Analyses (PRISMA).

**Figure 2 cimb-47-00544-f002:**
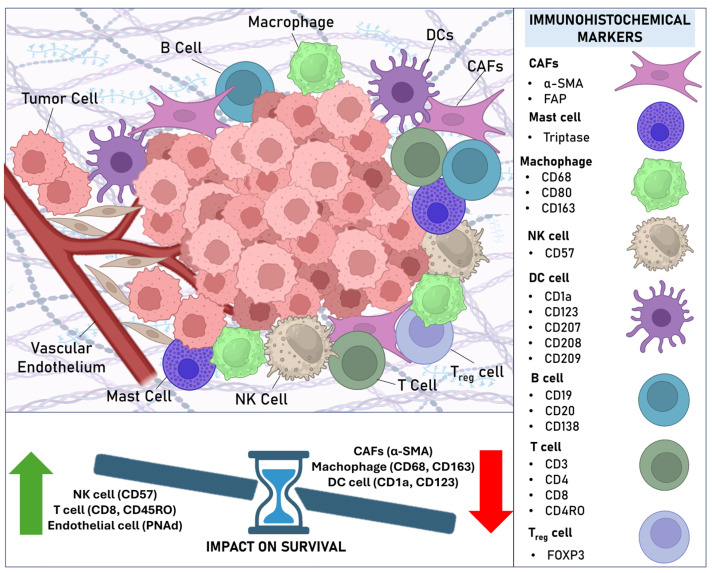
The tumor microenvironment in OSCC highlighting the immunohistochemical markers of stromal cell components analyzed in this systematic review and their impact on patient survival.

## Data Availability

The original contributions presented in this study are included in this article and Appendix A; further inquiries can be directed to the corresponding author.

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
