# Peer review of "Prognostic Value of the Immunohistochemical Detection of Cellular Components of the Tumor Microenvironment in Oral Squamous Cell Carcinoma: A Systematic Review"

_cimb, 2025, doi:10.3390/cimb47070544_

Round 1

Reviewer 1 Report

Comments and Suggestions for Authors

The manuscript “Prognostic value of the immunohistochemical detection of cellular components of the tumor microenvironment in oral squamous cell carcinoma: a systematic review” is a review article about the prognostic impact of cellular components of the tumor microenvironment in oral squamous cell carcinoma, analyzed through immunohistochemistry.

The manuscript is well-written and can be of interest for the readers due to the topic. The English is fine and does not require any substantial improvement; only few typos are present.

After a careful analysis of the manuscript, I evidenced some major concerns that must be addressed. I ask authors to address the following concerns in order to make the manuscript suitable for publication:

  1. Clearly state the novelty and specific contribution of this review compared to existing literature.
  2. The introduction section should be expanded providing a proper background of OSCC and role of microenvironment.
  3. A detailed table summarizing data reported in the review, including name of marker, samples size of the study, main findings and reference is mandatory.
  4. The review is quite short although the available literature on this topic is quite abundant. I request authors to include a paragraph before conclusions, in which they present recent biomarkers that have been studied in vitro and that could be interesting for further investigations. For instance, authors completely ignore a major regulator of cell and tumor microenvironment, the enzyme nicotinamide N-methyltransferase (NNMT) which can influence a number of enzymes involved in epigenetics, such as histone deacetylases sirtuins (PMID: 31043742; PMID: 36829935; PMID: 36817086). This enzyme has been demonstrated to be overexpressed in a variety of solid tumors including oral squamous cell carcinoma, where it contributes to the tumorigenicity and aggressiveness (PMID: 34827592). Notably, a number of NNMT inhibitors are already available, and their use is a promising strategy for targeted therapy in cancer (PMID: 34572571; PMID: 34704059; PMID: 34424711; PMID: 36104373). All these considerations should be included since they would let the manuscript totally cover the available literature.
  5. The authors should report the limitations of the cited studied and provide a critical evaluation of the reported findings.
  6. Please expand the conclusion to address translational challenges and future research directions

Author Response

Reviewer #1:

The manuscript “Prognostic value of the immunohistochemical detection of cellular components of the tumor microenvironment in oral squamous cell carcinoma: a systematic review” is a review article about the prognostic impact of cellular components of the tumor microenvironment in oral squamous cell carcinoma, analyzed through immunohistochemistry.

The manuscript is well-written and can be of interest for the readers due to the topic. The English is fine and does not require any substantial improvement; only few typos are present.

After a careful analysis of the manuscript, I evidenced some major concerns that must be addressed. I ask authors to address the following concerns in order to make the manuscript suitable for publication:

  1. Clearly state the novelty and specific contribution of this review compared to existing literature.

Response: We appreciate the suggestion. The relevance and contribution of the study have been incorporated into the final paragraph of the Introduction section.

  1. The introduction section should be expanded providing a proper background of OSCC and role of microenvironment.

Response: We appreciate the opportunity to improve the Introduction. Additional information regarding oral squamous cell carcinoma and the tumor microenvironment has been incorporated.

  1. A detailed table summarizing data reported in the review, including name of marker, samples size of the study, main findings and reference is mandatory.

Response: We apologize for the inconvenience. This information is provided in Supplementary Materials 4, 5, and 6. As our study included a total of 59 articles, and in order to comply with the journal’s guidelines and adhere to the PRISMA protocol recommendations for systematic reviews, we prepared comprehensive supplementary tables containing detailed information on the included studies. These tables present descriptive data, sample characterization, analyzed biomarkers, analytical methods, and main findings, as well as the methodological quality assessment and risk of bias for each study. In parallel, we aimed to synthesize the findings descriptively throughout the manuscript to complement the tables and ensure a clear and cohesive presentation of the results. We believe that, in doing so, the quality and transparency of data presentation have been preserved.

  1. The review is quite short although the available literature on this topic is quite abundant. I request authors to include a paragraph before conclusions, in which they present recent biomarkers that have been studied in vitro and that could be interesting for further investigations. For instance, authors completely ignore a major regulator of cell and tumor microenvironment, the enzyme nicotinamide N-methyltransferase (NNMT) which can influence a number of enzymes involved in epigenetics, such as histone deacetylases sirtuins (PMID: 31043742; PMID: 36829935; PMID: 36817086). This enzyme has been demonstrated to be overexpressed in a variety of solid tumors including oral squamous cell carcinoma, where it contributes to the tumorigenicity and aggressiveness (PMID: 34827592). Notably, a number of NNMT inhibitors are already available, and their use is a promising strategy for targeted therapy in cancer (PMID: 34572571; PMID: 34704059; PMID: 34424711; PMID: 36104373). All these considerations should be included since they would let the manuscript totally cover the available literature.

Response: We acknowledge the importance of novel biomarkers currently being investigated in vitro and recognize that we had not properly referenced these markers in the original version of the manuscript. In response to the reviewer’s suggestion, we added a section in the Results and Discussion entitled “Limitations and Challenges,” in which we included a paragraph addressing some of the emerging biomarkers that have shown promising results in oral squamous cell carcinoma or in head and neck squamous cell carcinoma.

  1. The authors should report the limitations of the cited studied and provide a critical evaluation of the reported findings.

Response: Alongside the new evidence on promising tumor microenvironment biomarkers, we also addressed their current limitations and how they might be overcome in future studies. Additionally, we highlighted the limitations of our own findings.

  1. Please expand the conclusion to address translational challenges and future research directions

Response: As requested, and in order to better address the reviewer’s considerations, we added a section on “Limitations and Challenges,” in which we discuss the translational barriers to clinical implementation and, based on these, propose future research directions.

Additional note: Due to the addition of new references within the main text, the numbering has been updated accordingly throughout the manuscript, as well as in the References section and the supplementary tables.

Reviewer 2 Report

Comments and Suggestions for Authors

The authors have conducted a systematic review on an important topic exploring the prognostic value of the immunohistochemical detection of cellular components of the tumor microenvironment in oral squamous cell carcinoma. Some concerns are listed below:

Inclusion criteria are missing.

Was there enough data to conduct Meta-Analysis?

More key results must be presented in Tables as a part of Results section, so the main findings are adequatly presented.

The manuscript lacks the authors’ expert opinion and critical perspective. A review article should go beyond summarizing existing literature by offering informed insights, highlighting knowledge gaps, and proposing future directions.

Add a section that critically discusses the limitations and challenges.

The previously published papers on similar subject must be included in Introduction.

Author Response

Reviewer #2:

The authors have conducted a systematic review on an important topic exploring the prognostic value of the immunohistochemical detection of cellular components of the tumor microenvironment in oral squamous cell carcinoma. Some concerns are listed below:

  • Inclusion criteria are missing.

Response: We appreciate the comment. We have clarified the inclusion criteria in the “Study Selection and Selection Criteria” section of the Materials and Methods.

  • Was there enough data to conduct Meta-Analysis?

Response: Unfortunately, it was not possible to perform a meta-analysis. Although this was the initial aim of the study, the high methodological heterogeneity among the included studies—both regarding biomarker evaluation criteria and analyzed outcomes—precluded quantitative data synthesis. Acknowledging this limitation, we explicitly highlighted it in the penultimate paragraph of the “Results and Discussion” section, emphasizing the need for methodological standardization in future research on this topic.

  • More key results must be presented in Tables as a part of Results section, so the main findings are adequatly presented.

Response: We appreciate the reviewer’s observation and understand the importance of presenting the main findings clearly and accessibly through tables in the Results section. After discussion among all authors, we decided to maintain the data in supplementary tables, considering the scope of the review and the large number of included studies (n = 59), which resulted in very extensive tables. To comply with the journal’s guidelines and adhere to the PRISMA protocol recommendations for systematic reviews, we prepared comprehensive supplementary tables that contain detailed information on the included studies, such as descriptive data, sample characterization, analyzed biomarkers, methods used, and main findings, as well as methodological quality assessment and risk of bias for each study. In parallel, we sought to synthesize the findings descriptively throughout the manuscript text to complement the tables and ensure a clear and cohesive presentation of the results. We believe that, in doing so, the quality and transparency of data presentation have been preserved.

  • The manuscript lacks the authors’ expert opinion and critical perspective. A review article should go beyond summarizing existing literature by offering informed insights, highlighting knowledge gaps, and proposing future directions.

Response: We appreciate the valuable comment and the opportunity to improve our manuscript. A similar remark was also made by Reviewer 1, which underscored the importance of incorporating a more critical analysis into the study. In response to these suggestions, and while respecting the word count limits set by the journal, we included current perspectives on emerging tumor microenvironment biomarkers identified through bioinformatic analyses, which hold potential as promising therapeutic targets for oral cancer. Additionally, we discussed the methodological limitations of these studies, as well as the challenges to be overcome for their clinical translation. By adding the section “Limitations and Challenges,” we also sought to emphasize the practical implications of the present review’s findings, highlighting knowledge gaps and proposing directions for future research. We believe that these modifications have contributed to deepening the critical analysis of the work, as requested.

  • Add a section that critically discusses the limitations and challenges.

Response: We appreciate the suggestion. The section has been added to the manuscript.

  • The previously published papers on similar subject must be included in Introduction.

Response: In accordance with the recommendation, we have expanded the Introduction.

Additional note: Due to the addition of new references within the main text, the numbering has been updated accordingly throughout the manuscript, as well as in the References section and the supplementary tables.

Round 2

Reviewer 1 Report

Comments and Suggestions for Authors
  1. I previously asked to clearly state the novelty and specific contribution of this review compared to existing literature. The authors state that “The relevance and contribution of the study have been incorporated into the final paragraph of the Introduction section” but in fact what they wrote does not highlight the novelty of this review.
  2. The concern regarding the background about OSCC has been properly addressed.
  3. The concern regarding point 3 has been properly addressed.
  4. The paragraph regarding the novel biomarkers did not take into account the available literature. Moreover, the reference 89 about NNMT is wrong and refers to a different work. The possible translational application of NNMT inhibitors has been ignored.
  5. The revised version of the manuscript should undergo English editing since some sentences are difficult to understand (e.g. Despite its comprehensive scope, several limitations were identified in the included studies; “recent TME biomarkers investigated” should be “recent investigated TME biomarkers” etc..).

Author Response

Reviewer #1:
1. I previously asked to clearly state the novelty and specific contribution of this
review compared to existing literature. The authors state that “The relevance and
contribution of the study have been incorporated into the final paragraph of the
Introduction section” but in fact what they wrote does not highlight the novelty of this
review.
Response: We have revised the final paragraph of the Introduction to clearly
highlight the novelty and specific contribution of our systematic review in comparison to
the existing literature. In this updated version, we explicitly address the current gap in the
literature and emphasize how our review provides a comprehensive and integrative
analysis of the prognostic relevance of various cellular components of the tumor
microenvironment in OSCC. We believe that the revised paragraph now fully meets the
reviewer’s request.
2. The concern regarding the background about OSCC has been properly addressed.
Response: Ok.
3. The concern regarding point 3 has been properly addressed.
Response: Ok.
4. The paragraph regarding the novel biomarkers did not take into account the
available literature. Moreover, the reference 89 about NNMT is wrong and refers to a
different work. The possible translational application of NNMT inhibitors has been
ignored.
Response: We thank the reviewer for this important observation. We have
replaced the previous paragraph with a revised and expanded version that appropriately
addresses the role of NNMT in OSCC. The citation has been corrected, and we
incorporated the most relevant references among those suggested, emphasizing not only
the prognostic significance of NNMT but also its translational potential through the use
of available inhibitors currently under investigation. Additionally, we enhanced the
discussion by including other novel biomarkers such as BANF1 and PES1, while
acknowledging that much of the current evidence is based on bioinformatic analyses and
requires further experimental validation.
5. The revised version of the manuscript should undergo English editing since some
sentences are difficult to understand (e.g. Despite its comprehensive scope, several
limitations were identified in the included studies; “recent TME biomarkers investigated”
should be “recent investigated TME biomarkers” etc..).
Response: We thank the reviewer for the valuable observation regarding the need
for English editing. All grammatical and stylistic issues identified have been carefully
reviewed and corrected throughout the revised manuscript. Problematic constructions
have been revised for improved clarity, accuracy, and fluency. The manuscript has now
been thoroughly edited to ensure consistency with scientific English writing standards.

Reviewer 2 Report

Comments and Suggestions for Authors

Thank you for your corrections to the manuscript. In its current form, the manuscript is suitable for acceptance and publication.

Author Response

I appreciate your careful review and thoughtful suggestions. 

Round 3

Reviewer 1 Report

Comments and Suggestions for Authors

The authors correctly addressed all the raised concerns therefore the manuscript can be accepted for publication.